# BTK Inhibitors and Other Targeted Therapies in Waldenström Macroglobulinemia

**Karan L. Chohan** [1] **and Prashant Kapoor** [2,*]

1 Department of Medicine, Mayo Clinic, Rochester, MN 55905, USA
2 Division of Hematology, Mayo Clinic, 200 First Street SW, Rochester, MN 55905, USA
* Correspondence: kapoor.prashant@mayo.edu; Tel.: +1-507-266-4800; Fax: +1-507-266-4972

**Abstract:** Waldenström macroglobulinemia (WM) is a rare, non-Hodgkin lymphoma that remains incurable. Rituximab, an anti-CD20 monoclonal antibody has been the cornerstone of treatment against WM, and its combination with an alkylator, bendamustine, achieves durable remission in treatment-naive patients with symptomatic WM. However, novel "druggable" targets that have been identified within the clonal lymphoplasmacytic cells in WM have resulted in a rapid development of targeted therapies in both the frontline and relapsed and refractory (R/R) settings. Several agents directed against the known targets have shown promising efficacy, with mostly manageable toxicities. The class of Bruton's tyrosine kinase (*BTK*) inhibitors has transformed the therapeutic landscape for patients with WM, given their convenient oral dosing and strong efficacy, with high rates of attainment of very good partial response (VGPR). The tolerability of the next-generation *BTK* inhibitors appears to be superior to that of the first-in-class agent, ibrutinib. Targeted therapies from other classes have also demonstrated efficacy in both single-agent and combination regimens. Inhibitors of proteasome BCL-2, mTOR and PI-3 kinase have demonstrated efficacy in WM. Emerging therapies under investigation will continue to further shape the management paradigm, especially in the R/R setting. These include bispecific antibodies, radiotherapeutic agents and chimeric antigen receptor T-cell (CART) cell therapies. This review outlines the current literature and future direction of targeted therapies in WM.

**Keywords:** lymphoplasmacytic lymphoma; IgM monoclonal gammopathy; *BTK* inhibitor; treatment

## 1. Introduction

Waldenström macroglobulinemia (WM) is a rare, non-Hodgkin lymphoma characterized by the clonal lymphoplasmacytic infiltration of the bone marrow, and circulation of the monoclonal immunoglobulin M (IgM) produced by the malignant cells [1]. Over the past two decades, the therapies against WM have evolved substantially, especially in the realm of targeted approaches [2]. The advent of Burton's tyrosine kinase inhibitors (*BTKi*) has transformed the therapeutic landscape for patients in both the relapsed and refractory (R/R), and more recently, the frontline setting. Other effective targeted therapies, such as proteasome inhibitors (PIs) have additionally demonstrated promising effects over the years, including the novel oral PIs such as ixazomib and oprozomib, and interestingly, more recently, the class of BCL-2 inhibitors (Figure 1). This review focuses on *BTKi* and other targeted therapies in WM and additionally touches upon novel targeted therapies that are under investigation.

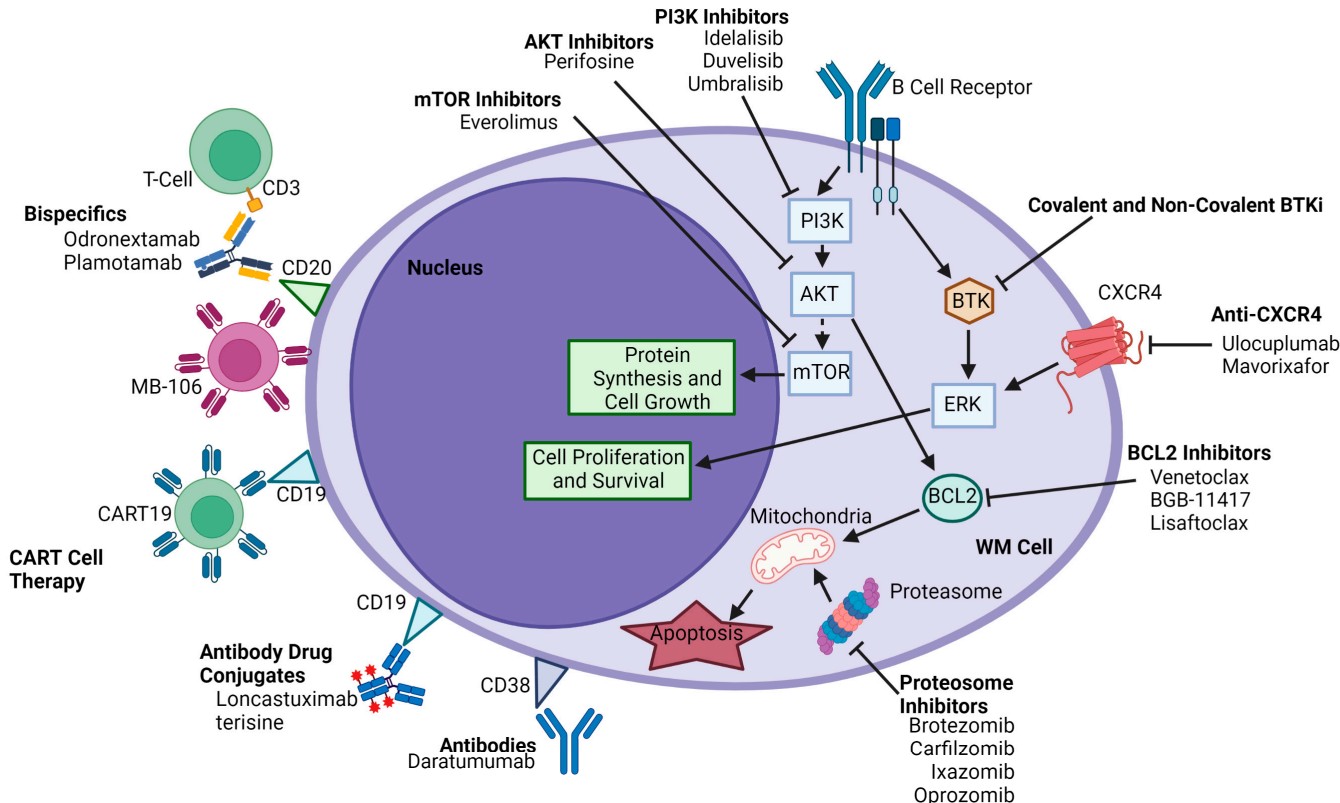

**Figure 1.** Mechanisms of action of select novel agents for the treatment of Waldenström macroglobulinemia (WM). AKT, protein kinase B; BCL2, B-cell lymphoma 2; *BTK*, Bruton tyrosine kinase; *BTKi*, *BTK* inhibitor; CART, chimeric antigen receptor T; CART19, anti-CD19 CART; CD, cluster of differentiation; *CXCR4*, C-X-C chemokine receptor type 4; ERK, extracellular signal-regulated kinase; mTOR, mammalian target of rapamycin; PI3K, phosphoinositide 3-kinase.

## 2. Genomic Landscape of WM

Based on greater understanding of the genomic landscape of WM through the employment of more sophisticated laboratory tools, including next-generation sequencing and allele-specific PCR assays, the value of assessing the genotype (mutational status) of the patients in therapy selection and prognostication is increasingly being recognized with this malignancy. Common mutations in WM involve the myeloid differentiation primary response 88 (*MYD88*) gene, occurring in over 90% of WM cases, C-X-C chemokine receptor type 4 (*CXCR4*), AT-rich interactive domain 1A (*ARID1A*) and cluster of differentiation (CD)79 [3,4]. Among the *MYD88* mutations, the vast majority of changes result in leucine to proline substitution (L265P), and non-L265P mutations such as S219C, M232T and S243N are infrequently encountered, comprising 1–2% of mutations in the *MYD88* gene [5,6]. The patients with $MYD88^{\text{wild-type(WT)}}$ genotype often lack response to ibrutinib therapy [7]. *MYD88* is an important adaptor protein which leads to downstream activation of the nuclear factor kappa B (NFKB) pathway through interaction with the toll-like receptor (TLR), interleukin (IL-1) receptor families and recruitment of other proteins (Figure 1) [8]. *CXCR4* is a G protein-coupled chemokine receptor, leading to activation of multiple downstream pathways. In WM, several mutations of *CXCR4* have been identified in up to 40% of patients [4]. Chromosome 6q deletion (del6q), encountered in about half of WM patients, results in the loss of important negative regulators of the NFKB pathway [9–11].

## 3. Burton Tyrosine Kinase Inhibitors

The enzyme *BTK* is expressed in B cells and plays a crucial role in the cellular signaling pathways that regulate cell survival and proliferation [12]. *BTK* is activated in WM secondary to multiple genomic alterations [13]. The inhibition of *BTK* by targeted agents has changed the paradigm of WM management, especially in the R/R setting (Figure 2). Covalent *BTKi* such as ibrutinib, zanubrutinib, and acalabrutinib, orelabrutinib and tirabrutinib irreversibly bind to *BTK* at the cysteine 481 (C481) active site. This results in the blocking of B-cell receptor (BCR) signaling. Unfortunately, these covalent agents can be limited by their requirement for daily continuous use, intolerance caused by off-target inhibition of other kinases, acquisition of mutations, including *BTK*$^{Cys481}$ mutations that may lead to disease progression [14–16]. The C481S mutation in the *BTK* gene is frequently encountered in patients who develop resistance to ibrutinib and has been demonstrated to restore BCR signaling through extracellular signal-regulated kinase 1/2 (ERK1/2) reactivation [15]. This leads to the release of pro-survival and inflammatory cytokines, including interleukin (IL) 6 and IL 10, by the *BTK*$^{Cys481Ser}$ harboring WM cells, conferring ibrutinib resistance even among the *BTK*$^{WT}$ *MYD88*-mutated WM cells [15]. Much of the toxicities of *BTKi* therapies are mediated through off-target inhibition of kinases (Figure 2) such as IL-2-inducible T-cell kinase (ITK), tyrosine-protein kinase (TEC), and endothelial growth factor receptor (EGFR). The toxicity profile of individual *BTKi* agents is mediated through their selectivity for *BTK*, and the degree of their off-target kinase binding [17]. The individual safety profile of current *BTKi* therapies is discussed below; however, major toxicities include atrial fibrillation, hypertension, ventricular arrhythmias, bleeding risk, infection, myalgias, arthralgias, diarrhea and cytopenias [7,18]. Recently, non-covalent *BTKi*, including pirtobrutinib and nemtabrutinib, which are structurally different from the covalent *BTKi*, have been investigated, with encouraging efficacy demonstrated even in patients intolerant or refractory to covalent *BTKis*, offering a viable alternative in those scenarios [19]. Non-covalent agents do not directly bind the C481 on *BTK*, thus circumventing the resistance acquired due to *BTK*$^{Cys481}$ mutations that develops on continuous covalent *BTKi*-based regimens [19].

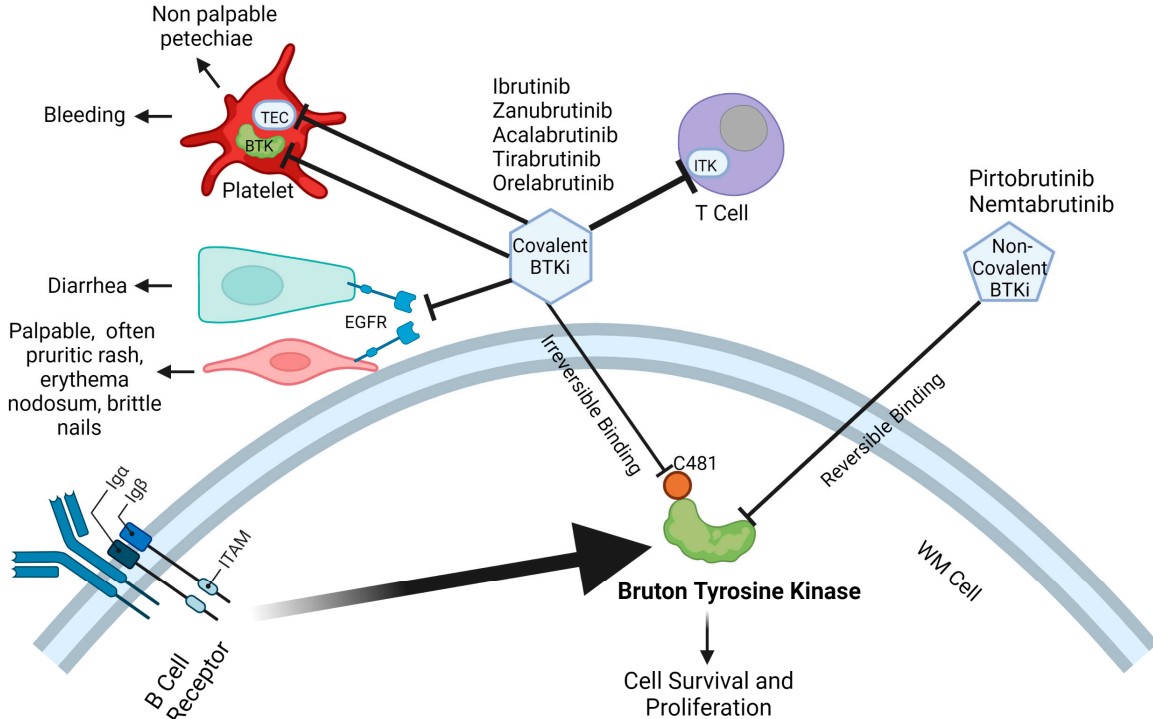

**Figure 2.** A comparison of covalent and non-covalent Bruton tyrosine kinase (*BTK*) inhibitors. C481, cysteine-481. *BTK*, Bruton tyrosine kinase; ITK, IL-2 inducible T-cell kinase; TEC, tyrosine protein kinase.

### 3.1. Covalent Bruton Tyrosine Kinase Inhibitors

#### 3.1.1. Ibrutinib

Ibrutinib, the first-in-class covalent *BTKi*, was approved in 2015 after the pivotal trial by Treon et al. (Table 1) revealed high rates of clinical response and improvement in anemia in the R/R setting [7]. Its mechanism of action relies upon the irreversible binding to the C481 site on *BTK*. Thus, ibrutinib prevents downstream signaling and activation through blocking phosphorylation. The seminal phase 1 study by Advani et al. involving patients with various B cell malignancies demonstrated an encouraging overall response in three of four patients with R/R WM, treated with oral ibrutinib monotherapy [20]. This led to subsequent phase 2 and 3 trials investigating ibrutinib in patients with R/R disease and in the frontline settings [7,18]. In the Treon study involving 63 R/R patients, at approximately 4 years of follow-up, an overall response rate (ORR) of 91% and a major response rate (MRR) of 73% were observed [7]. Additionally, the 2-year overall survival (OS) and progression-free survival (PFS) rates in the R/R were impressive at 95% and 69%, respectively. Interestingly, the study demonstrated higher response rates among patients with $MYD88^{L265P}CXCR4^{WT}$ signature as compared to those with $MYD88^{WT}CXCR4^{WT}$ or $MYD88^{WT}CXCR4^{WHIM}$ genotype [7]. Subsequent analysis revealed that ibrutinib monotherapy was ineffective in patients with a true $MYD88^{WT}$ signature and failed to achieve even a partial remission (PR) in that cohort, indicating the need to assess patients' $MYD88^{L265P}$ genotype status prior to embarking on ibrutinib monotherapy. Some experts recommend reflex sequencing for the non-L265P *MYD88* mutations among patients initially found to harbor $MYD88^{WT}$ signature, as the presence of non-L265P *MYD88* mutations is predictive of response to ibrutinib in such patients who would otherwise have been miscategorized as $MYD88^{WT}$. Long-term follow-up results from this phase 2 trial at a median of 59 months revealed an ORR and MRR of 91% and 79%, respectively, with a 5-year OS that remained at a high of 87% for all patients [21]. The 5-year PFS rate was not reached for the entire cohort, a result indicative of durable remission. Not surprisingly, the strong efficacy (ORR: 100%, MRR: 83%) of single-agent ibrutinib extended even among patients with $MYD88^{L265P}$ mutation who were treatment naïve (TN) in another phase 2 trial [22]. The time to response was rapid, with a median of 4 weeks, and 18-month PFS and OS rates were 92% and 100%, respectively, in this small phase 2 trial involving 30 patients. Owing to multiple studies confirming the promising, but varying efficacy of continuous ibrutinib therapy in different patient subpopulations, regimens incorporating ibrutinib are recently being explored in combination with other agents, with attempts to limit the duration of therapy and to improve upon the efficacy of ibrutinib monotherapy. The randomized, placebo-controlled phase 3 iNNOVATE trial investigated 150 TN and R/R patients with WM who were randomly assigned to receive an oral placebo with rituximab (plc/R) or ibrutinib with rituximab (ibr/R) [23]. At 30 months of follow-up, the response rates were higher as predicted in the ibr/R arm compared to pcb/R (Ibr/R—ORR: 92%, MRR: 72% vs. Pcb/R—ORR: 47%, MRR: 32%), with improved 30-month PFS outcomes but not OS outcomes (Ibr/R PFS: 82%, OS: 94% vs. Pcb/R PFS: 28%, OS: 92%). This doublet was touted as potentially overcoming the poor efficacy of ibrutinib monotherapy in the $MYD88^{WT}$ population, although its value in such a patient population could not be accurately ascertained in the iNNOVATE trial, given the use of a suboptimal non-PCR-based assay with lower sensitivity, likely leading to false negative results, and in turn, miscategorization of patients as $MYD88^{WT}$. Moreover, the study did not have an ibrutinib monotherapy arm for comparison, and Pcb/R, albeit frequently utilized in the real world for WM, is a suboptimal control. The final results from the iNNOVATE trial at a median follow-up of 50 months revealed a PFS benefit with Ibr/R (median not reached), compared with Pcb/R (median: 20.3 months) [24]. Furthermore, the median time to next treatment was lower in the Pcb/R group (18 months) compared to the Ibr/R group. Overall, the ongoing superiority of ibr/rituximab was demonstrated. The rates of rituximab flare and infusion related reaction were unsurprisingly less in the combination arm. Efficacy seemed to be equivalent in the TN and R/R population. Despite good efficacy demonstrated in the multiple trials outlined above, several adverse events of

*BTKi* occur, predominantly due to off-target inhibition of other kinases. Ibrutinib has been associated commonly with gastrointestinal disturbance, but there exists a significant and concerning risk of developing cardiovascular toxicity, including cardiac arrhythmias and hypertension and bleeding [25]. In a systematic review and pooled analysis, comparing ibrutinib therapy to placebo, increased rates of developing serious atrial fibrillation/flutter (3.0% vs. 0.8%, $p = 0.003$), all grade atrial fibrillation/flutter (8.2% vs. 0.9%, $p < 0.001$) and bleeding (3.7% vs. 2.1%, $p = 0.07$) were shown [25]. Life-threatening ventricular arrhythmias have also been encountered, dampening enthusiasm for its use in countries where alternative *BTKi* with a superior cardiovascular profile are available.

**Table 1.** Select key clinical trials of targeted therapies in the treatment of WM.

| Class | Agent (Ref) | No of WM Pts | Response | Overall Survival | Progression-Free Survival | Notable Adverse Events * | Comments |
|---|---|---|---|---|---|---|---|
| *BTK* Inhibitors | Ibrutinib [7] | 63 R/R | ORR: 91% MRR: 73% | 2-year OS: 95% | 2-year PFS: 69% | Atrial fibrillation: 5% (2%) Bleeding: 2% (0) Neutropenia: 22% (15%) Thrombocytopenia: 14% (13%) | Response rate higher in $MYD88^{\text{L265P}}CXCR4^{\text{WT}}$ vs. $MYD88^{\text{WT}}CXCR4^{\text{WT}}$ or $MYD88^{\text{WT}}CXCR4^{\text{WHIM}}$ |
| | Ibrutinib [22] | 30 TN $MYD88^{\text{MUT}}$ | ORR: 100% MRR: 83% | 18-mo OS: 100% | 18-mo PFS: 92% | Atrial fibrillation 7% (0) Hypertension: 13% (7%) Bruising: 7% (0) Neutropenia: 7% (0) | No patients with $MYD88^{\text{WT}}$ status were enrolled. |
| | Ibrutinib/rituximab vs. placebo/rituximab (23) | 150 | ORR: Ibr/R 92% vs. Pcb/R 47% MRR: Ibr/R 72% vs. Pcb/R 32% | 30-mo OS: Ibr/R 94% vs. Pcb/R 92% | 30-mo PFS: Ibr/R 82% vs. Pcb/R 28% | Atrial fibrillation (grade 3 or higher): 12% Ibr/R vs. 1% Pcb/R Hypertension (grade 3 or higher): 13% Ibr/R vs. 4% Pcb/R Major hemorrhage (all): 4% both arms | Patients in the control arm could cross over to the IR arm upon progression. Reduced rates of rituximab related infusion reactions were noted with concomitant ibrutinib administration. |
| | Ibrutinib vs. Zanubrutinib [26] | 164 R/R, 37 TN (Zanu: 102 Ibr: 99) | ORR: Zanu 94% vs. Ibr 93% MRR: Zanu 77% vs. Ibr 78% | 18-mo OS: Zanu 97% vs. Ibr 93% | 18-mo PFS: Zanu 85% vs. Ibr 84% | Atrial fibrillation (grade 3 or higher): 0% Zanu vs. 4% Ibr Hypertension (grade 3 or higher): 6% Zanu vs. 11% Ibr Neutropenia (grade 3 or higher): 20% Zanu vs. 8% Ibr | VGPR rates were numerically higher with zanubrutinib. |
| | Zanubrutinib [27] | 23 R/R 5 TN | ORR: 81% MRR: 50% | 18-mo OS: 88% | 18-mo PFS: 68% | All hemorrhage: 39% (7%) Hypertension: 11% (11%) Neutropenia: 18% (11%) Atrial fibrillation: 4% (0%) | All pts $MYD88^{\text{WT}}$ |
| | Acalabrutinib [28] | 14 TN, 92 R/R | ORR: 93% TN, 92% R/R MRR: 79% TN, 78% R/R | 24-mo OS: 92% TN, 89% R/R | 24-mo PFS: 90% TN, 82% R/R | Bleeding: 58% (3%) Atrial fibrillation: 5% (1%) Pneumonia: 9% (7%) Hypertension: 5% (3%) | Response rates were similar in both TN and R/R patients |
| | Tirabrutinib [29] | 18 TN, 9 R/R | ORR: 96% MRR: 89% | NR | NR | Neutropenia: 30% (11%) Lymphopenia 11% (11%) Rash: 44% (NA) | Short follow-up duration but therapy efficacy appears comparable in TN and R/R patients. Rash is a major toxicity. |
| | Pirtobrutinib [30] | 80 R/R | ORR: 84% MRR: 73% | 19 mo in prior *BTKi* patients | NR in prior *BTKi* patients | Bruising: 24% (0%) Neutropenia 24% (20%) Atrial fibrillation/flutter 3% (1%) Hypertension 9% (2%) | High rates of response seen in patients with previous *BTKi* treatment |

**Table 1.** *Cont.*

| Class | Agent (Ref) | No of WM Pts | Response | Overall Survival | Progression-Free Survival | Notable Adverse Events * | Comments |
|---|---|---|---|---|---|---|---|
| | Orelabrutinib [31] | 66 R/R | ORR: 89% MRR: 81% | 12-mo OS: 94% | 12-mo PFS: 89% | Neutropenia: 19% (11%) Thrombocytopenia: 28% (6%) Pneumonia 4% (4%) Hepatitis B reactivation: 2% (2%) | Hepatitis B reactivation is a major safety concern needing further exploration. |
| BCL2 Inhibitors | Venetoclax [32] | 32 R/R | ORR: 84% MRR: 81% | 30-mo OS: 100% | 24-mo PFS: 80% | Neutropenia 53% (45%) Anemia: 25% (3%) Tumor lysis syndrome: 3% (3%) | Progression noted in 10 patients within a year of completion of 24-mo fixed duration treatment |
| | Bortezomib [33] | 10 R/R | ORR: 80% PR: 60% | NR | NR | Thrombocytopenia: 40% (20%) Fatigue: 70% (20%) Peripheral neuropathy: 30% (20%) | Peripheral neuropathy was a major concern limiting applicability |
| | Bortezomib [34] | 12 TN, 15 R/R | ORR: 26% MRR: 26% | OS: NA | PFS: 16.3 mos | Thrombocytopenia: 70% (30%) Neutropenia: 67% (19%) Peripheral Neuropathy: 74% (19%) | Peripheral neuropathy was a major concern limiting applicability |
| PI | Carfilzomib-rituximab-dexamethasone (CaRD) [35] | 33 TN | ORR: 87% MRR: 68% | 15.4 median follow-up: OS 100% | 15.4 median follow-up: PFS 65%, | Infusion reaction: 23% (0%) Rash: 29% (0%) Thrombocytopenia: 3% (0%) Hyperlipasemia: 56% (16%) | Cardiopulmonary toxicity remains a major concern with carfilzomib-based regimens. |
| | Ixazomib-dexamethasone-rituximab (IDR) [36] | 26 TN | ORR: 96% MRR: 77% | OS 100% at 52-month follow-up | Median PFS: 40 months | Insomnia: 27% (8%) Rash: 27% (8%) Infusion reaction: 39% (19%) | Grade 3 neuropathy was noted in only 1 patient |
| | Oprozomib [37] | 71 WM and MM | ORR: 71% 2/7 schedule 50% 5/14 schedule MRR: 50% 2/7 schedule, 29% 5/14 schedule | OS: NA | Median PFS: 6.1 mos 2/7 schedule, 3.7 mos 5/14 schedule | Phase II Events for WM-Diarrhea: 2/7 schedule 93% (27%), 5/14 schedule 71% (6%) Sepsis: 2/7 schedule 13% (13%), 5/14 schedule 0% Decreased Platelets: 2/7 schedule 40% (7%), 5/14 18% (0%) | Therapy limited by grade 3 or greater gastrointestinal and hematologic events |
| PI3Ki | Idelalisib and Obinutuzumab [38] | 48 R/R | ORR: 71% MRR: 65% | 12-mo OS: 90% | 12-mo PFS: 55% | Neutropenia: 19% (16%) Diarrhea: 9% (5%) Hepatic Toxicity: 9% (7%) | Hepatic toxicity limited clinical applicability |

* All grade% (≥grade 3%) unless otherwise indicated. *BTK*, Burton's tyrosine kinase; Ibr, ibrutinib; mo, months; MRR, major response rate; NA, not available; NR, not reached; ORR, objective response rate; OS, overall survival; PFS, progression-free survival; PI, proteasome inhibitors, PI3Ki, phosphatidylinositol 3-kinase inhibitor; Plb, placebo; PR, partial response; Pts, patients; R, rituximab; Ref, reference; R/R, relapsed/refractory; TN, treatment-naïve; Zanu, zanubrutinib.

### 3.1.2. Zanubrutinib

Subsequent to the approval of ibrutinib, second-generation *BTKi*, including zanubrutinib and acalabrutinib, have demonstrated equivalent or somewhat superior efficacy, and zanubrutinib has received approval for use in WM by the regulatory bodies (Table 1). These second-generation agents were developed with the aim of reducing off-target effects on other kinases [39]. In Cohort 1 of the phase 3 ASPEN trial, zanubrutinib was compared to ibrutinib monotherapy in 201 patients with $MYD88^{L265P}$ mutation who were either previously treated (*n* = 164), or TN (*n* = 37). Patients who were TN were required to be unsuitable for standard immunotherapy based on comorbidities or risk factors at the time of study entry [26]. The patients were randomly assigned to receive either of the two *BTKi*. However, complete response (CR) was not observed in either arm. A higher proportion of patients achieved VGPR with zanubrutinib (28%) compared to ibrutinib (19%), although this difference was not statistically significant in the initial analysis, indicating that the trial

had failed to meet its primary endpoint. Furthermore, the major response rates (MRR) were similar in both arms (zanubrutinib: 77%; ibrutinib: 78%). In the zanubrutinib arm, there was a lower rate of all-grade atrial fibrillation (2% vs. 15%, $p < 0.005$), but neutropenia occurred more frequently (29% vs. 13%) as compared to ibrutinib. Interestingly, similar rates of grade 3 or greater infection were observed in the 2 groups. Therefore, the primary analysis of this trial showed that the efficacy of both agents was similar, but superior tolerability of zanubrutinib as compared to ibrutinib, lending the use of the former as the preferred approach. Long-term follow-up results were recently presented where the rates of VGPR were higher in Cohort 1 comparing zanubrutinib to ibrutinib (36% vs. 22%, $p = 0.02$) [40]. Along with the improved efficacy, zanubrutinib continued to be associated with lower rates of atrial fibrillation/flutter (8% vs. 24%), hypertension (15% vs. 26%), and discontinuation due to AEs (9% vs. 19%) [41]. Exposure-adjusted incidence rates were additionally lower with zanubrutinib for both atrial fibrillation/flutter (0.2 vs. 0.8 persons per 100 person-months, $p < 0.05$) and hypertension (0.5 vs. 1.0 persons per 100 person-months, $p < 0.05$) as compared to ibrutinib. In a non-randomized portion of the ASPEN trial, a second cohort, Cohort 2, exclusively involving patients with $MYD88^{WT}$ genotype, the ASPEN study also evaluated in 28 patients on zanubrutinib (5 TN, 23 R/R) monotherapy [27]. Here, a VGPR/CR rate of 31% was observed, with one patient achieving a CR. In Cohort 2, atrial fibrillation was observed in 4% of patients, treatment-emergent hypertension occurred in 11% of patients, with major bleeding, defined as grade 3 or higher hemorrhage and CNS bleeding of any grade, in 7%, indicating that the toxicities associated with zanubrutinib were not trivial either.

### 3.1.3. Acalabrutinib

Acalabrutinib was investigated in a single-arm phase 2 multicenter study of 106 patients with WM, including 14 TN and 92 R/R patients [28]. Here, oral acalabrutinib resulted in an ORR of 93% with an MRR of 78%. The response rates were similar in both TN and R/R patients, with an MRR of 79% and 78%, respectively. Atrial fibrillation occurred in 5% of patients, and there was an overall low rate of AEs, including neutropenia (16%), anemia (5%) and pneumonia (7%). Headache was a notable AE. Although acalabrutinib has not been compared head-to-head with the first generation *BTKi*, ibrutinib among patients with WM, in the ELEVATE RR trial, ibrutinib was compared with acalabrutinib in 533 patients with R/R CLL who were randomly assigned 1:1 to either agent [42]. Acalabrutinib was determined to be non-inferior with a median PFS of 38.4 months in both arms. However, similar to the ASPEN trial, the rates of all-grade atrial fibrillation and flutter were found to be higher with ibrutinib (16.0% vs. 9.4%. $p = 0.002$) as was the treatment discontinuation rate due to AEs (21% vs. 15%) compared to the acalabrutinib arm [42].

### 3.2. Other Emerging Covalent Bruton Tyrosine Kinase Inhibitors

Currently, there are several ongoing trials assessing combination regimens with other second-generation covalent, irreversible *BTKi*, including tirabrutinib, orelabrutinib and TG-1701 (Table 1) [29,31]. Tirabrutinib was evaluated in an multi-center, open-label, single-arm phase 2 trial that showed an ORR of 94% (assessed by an independent review committee) in TN patients (*n* = 18), and 100% (88.9% as assessed by an independent review committee) in R/R patients (*n* = 9) [29]. The most common AEs were rash (44%), neutropenia (26%) and leukopenia (22%), with grade 3 or higher AEs of neutropenia (11%), leukopenia (7%) and lymphopenia (11%). The PFS and OS rates during the short follow-up were 100% at 6 months both in the TN and R/R groups. These observations led to tirabrutinib being the first *BTKi* to be approved for the treatment of patients with TN or R/R WM and LPL in Japan. Orelabrutinib was investigated in 66 R/R WM patients in whom at 16.4 months of median follow-up, an MRR of 81% and ORR of 89% were observed [31]. The most common grade 3 or higher AEs included neutropenia (11%), thrombocytopenia (6%) and pneumonia (4%). Patients who were hepatitis B surface antigen positive were excluded from the study. Concerningly, one patient had grade 5 hepatitis B reactivation, which

was considered to be treatment-associated. Thus, it is important to note that reactivation of hepatitis B may occur with *BTKi* therapy, and appropriate screening prior to therapy and treatment should be employed. Currently, there is an ongoing trial assessing TG-1701 (NCT03671590) in B-cell malignancies. Overall, how these "me too" covalent *BTKi* will be able to supplement zanubrutinib in the absence of head-to-head trials demonstrating superior efficacy or toxicity remains to be seen.

### 3.3. Non-Covalent Bruton Tyrosine Kinase Inhibitors

3.3.1. Pirtobrutinib

Pirtobrutinib is a reversible inhibitor of *BTK* demonstrating efficacy in patients with $BTK^{WT}$ as well as $BTK^{C481S}$, or $BTK^{C481R}$ mutations (Table 1) [30,43]. Pirtobrutinib was investigated in the BRUIN study, a phase 1/2, multicenter, open-label trial in patients with R/R B-cell malignancies (NCT03740529), including CLL, MCL and WM. The initial results were published by Mato et al. in 2021, and a subsequent subset analysis assessing 80 patients with WM was recently presented at the American Society of Hematology (ASH) 2022 Annual Meeting [44]. Seventeen patients were *BTKi* naive and 63 were previously exposed to a covalent *BTKi* but either relapsed on it (67%) or discontinued the covalent *BTKi* due intolerance (33%). In this subset analysis, 91% of patients received the recommended phase 2 dose of 200 mg once daily. The most common all grade (grade $\geq$ 3) AEs observed in all patients with B-cell malignancies treated (*n* = 725) included fatigue in 29% (2%), diarrhea in 24% (1%) and bruising in 24% (0%), with neutropenia observed in 24% (20%), hypertension 9% (2%), atrial fibrillation/flutter in 2.8% (1%), rash in 13% (1%), arthralgia in 14% (1%) of patients. A high rate of response was seen in patients who had previously received a *BTKi* (MRR: 67%; CR + VGPR: 24%) and even higher among the *BTKi*-naïve population (MRR 88%; CR + VGPR 29%). The median PFS among the 63 patients with a prior exposure to a covalent *BTKi* was 19.4 months and OS was not reached among these patients (18-month OS rate 81.7%). The median follow-up for PFS and OS was 14 months and 16 months, respectively [44]. The data differentiating between patients with previous *BTK* exposure and subsequent discontinuation due to intolerability, compared to *BTK* refractory disease have not been currently parsed. Pirtobrutinib is not approved for WM as of this writing but has gained approval in January 2023 for patients with R/R MCL who have received at least two prior lines of therapy, including a covalent *BTKi*. The superiority of pirtobrutinib over covalent *BTKi* has not been established, but the encouraging data, particularly with respect to its safety, are noteworthy, and its availability in the US now will likely lead to increased off-label uses, including WM, particularly among patients intolerant to covalent *BTKi* or those who tolerated a covalent *BTKi* but ultimately relapsed on it, making it a valuable addition to the therapeutic armamentarium.

3.3.2. Nemtabrutinib

Nemtabrutinib, another reversible *BTK* inhibitor, has been shown to bind both WT and C481-mutated *BTK* (Table 1) [45]. Currently, it is being assessed in the phase 1/2 BELLWAVE-001 study (NCT03162536) and the preliminary results of 112 patients with B-cell NHL, including 6 patients with WM, were recently presented at the ASH 2022 Annual Meeting [45]. Nemtabrutinib demonstrated an ORR of 56% in patients with CLL and small cell lymphoma (SLL), with two patients achieving CR, 15 PR and 15 PR and residual lymphocytosis. Currently, the data on treatment efficacy in patients with WM are not available. The most common grade 3 or 4 AEs included neutrophilia (17%), thrombocytopenia (5%), and lymphocytosis (5%). Further evaluation of non-covalent reversible *BTKi* therapies and where they may appropriately fit in the treatment algorithm of WM is necessary and will ultimately prove very useful. Based on the available data, we believe that it is quite reasonable to use it off-label in R/R WM patients, refractory or intolerant to a covalent *BTKi*.

### 3.4. Further Considerations with BTKi

One of the concerns with abrupt discontinuation or interruption of *BTKi* therapy is IgM rebound [46]. This can result in a hyperactive immune state and has been observed in many patients post-ibrutinib discontinuation. Re-initiation generally results in the restoration of low IgM levels [47]. Future studies must investigate the optimal approach to therapy interruption, discontinuation and transition to prevent this outcome. The optimal approach to switching between classes of *BTKi* therapies is, additionally, not well known. A trial is currently investigating the utility of zanubrutinib in patients with intolerance to other *BTKi*, excluding those who progressed with prior *BTKi* therapy (NCT04116437). At ASH 2021, Shadman et al. presented the initial results, primarily from Cohort 1, which investigated the efficacy and safety profile of zanubrutinib in patients with B cell malignancies who were intolerant to only ibrutinib therapy [48]. Among 57 patients from this cohort, including nine patients with WM, 60% of patients had no recurrence of any ibrutinib intolerance events with the switch to zanubrutinib therapy. Additionally, no ibrutinib intolerance events recurred at higher severity with the change to zanubrutinib which resulted in an ORR of 63%, and stable disease or better in 95% of patients in Cohort 1. Recently, at ASH 2022, as part of Cohort 2 of this study, Shadman et al. presented data involving 17 patients, including three patients with WM, who received zanubrutinib due to acalabrutinib intolerance [49]. The study found that the majority of patients (65%) did not experience any recurrence of prior acalabrutinib intolerance events. Furthermore, efficacy of zanubrutinib was observed, with 93% achieving at least stable disease, and 64% exhibiting further deepening of response. Overall, these results demonstrate that patients who are intolerant to ibrutinib or acalabrutinib can respond to zanubrutinib, with the potential for avoidance of the toxicities/intolerance encountered with other less specific covalent *BTKi*. Recently, Sarosiek et al. investigated the impact of dose reduction of ibrutinib in the treatment of WM and the impact on patient outcomes in a retrospective study [50]. Of 353 WM patients treated with ibrutinib, 27% required a dose reduction due to AEs. Most patients had improvement or resolution of AEs (65%) after initial reduction. Furthermore, maintenance of hematologic response after reduction was demonstrated. Overall, a better understanding of the impact of dose modifications, and switching between different *BTKi* will provide improved guidance in treatment algorithms on the management of toxicities and how best to reduce the impact of AEs while maintaining optimal response.

## 4. BCL-2 Inhibitors

The anti-apoptotic BCL-2 family proteins are overexpressed in multiple malignancies, including WM [51]. Overexpression of BCL-2 confers prosurvival signaling. Oblimersen sodium is an antisense oligonucleotide for the first six codons of the BCL-2 reading frame which was investigated in a phase 1/2 trial in patients with symptomatic R/R WM [52]. The results of nine patients treated at two different dose levels were reported. Here, at dose level 1 ($n = 6$), grade 3 or higher hematologic toxicities were observed in the majority of patients ($n = 5$), and 3 patients had severe neutropenia. One patient was found to have a partial response [52]. Venetoclax is a highly selective BCL-2 inhibitor that was prospectively assessed in a phase 2 trial in R/R WM, following notable activity in patients with WM in the initial phase 1 study (Table 1) [53]. The phase 1 study investigated venetoclax in 106 patients with R/R NHL, including a very small subset ($n = 4$) patients with WM. Here, an ORR of 44% was observed in the entire cohort, with all 4 patients with WM achieving a PR [54]. However, three patients had laboratory tumor lysis syndrome (TLS) documented. The most common grade 3 or 4 AEs included anemia (15%), neutropenia (11%), and thrombocytopenia (9%). In a multicenter small phase 2 study involving 32 evaluable patients (16 with previous *BTKi* treatment, all *MYD88*$^{L265P}$-mutated, 17 *CXCR4* mutations), with a median follow-up of 33 months, high rates of response were observed (ORR: 84%, MRR: 81%, VGPR: 19%) with 24-month fixed-duration venetoclax monotherapy; however, CR was not reported with venetoclax as well [32]. The median PFS was observed to be 30 months, with a sizable proportion of patients progressing within 6 months of completing the two-year

therapy, making a case for continuous indefinite therapy rather than abrupt discontinuation, even among the responders. In terms of safety, 94% of patients experienced grade 2 or higher AEs, with neutropenia ($n$ = 6) and febrile neutropenia ($n$ = 1) being the only grade 4 AEs that were observed. Additionally, there was one case of grade 3 TLS with laboratory findings, but without any clinical sequelae. Overall, the remarkable safety and efficacy of venetoclax along with the ease of administration with once-daily oral route after the initial ramp up to reduce the rate of TLS, demonstrates that future investigation is warranted with this promising agent in earlier lines of treatment. Currently, venetoclax is being explored in combination regimens (NCT05099471). A phase 2 study evaluated a fixed-duration (2 years) combination of venetoclax with ibrutinib in 45 (of a total of 50 planned) TN patients with WM [55]. Here, high rates of response were seen with an MRR of 93% and VGPR of 40%; however, an unexpectedly high rate of ventricular arrhythmias found in 9% of the study population prompted premature closure of this study, with all patients being permanently taken off the oral doublet. Considering the development of fatal ventricular arrhythmias with the ibrutinib-venetoclax combination in the aforementioned study, an ongoing SWOG phase 2 randomized study (NCT04840602), evaluating ibrutinib and rituximab +/− venetoclax among patients with previously untreated WM/lymphoplasmacytic lymphoma has been suspended for the reassessment of the venetoclax-ibrutinib-based combination under investigation. A European trial yet to begin recruitment, ViWA-1 (NCT05099471), will investigate the combination of venetoclax-rituximab against DRC (control) in patients with TN WM.

BGB-11417 is a novel BCL-2 inhibitor that is currently being investigated in mature B-cell malignancies (NCT04277637), including WM. BGB-11417 is highly selective for BCL-2 and has demonstrated a higher potency than venetoclax in biochemical assays (>10:1 venetoclax) [56]. Preliminary ongoing phase 1 study results assessing BGB-11417 monotherapy and in combination with zanubrutinib among 45 patients with B-cell malignancies (34 monotherapy; 11 combination) was presented at the ASH 2022 Annual Meeting [57]. From the data presented, 6 patients with WM were treated with BGB-11417 monotherapy. Dose escalation to 640 mg was completed for NHL monotherapy. The most common treatment-related adverse events (TRAEs) reported were nausea (38%), fatigue (24%), and dizziness (21%), with the most common grade 3 or higher TRAE being neutropenia (12%). A significant number of patients (monotherapy: 25; combination: 2) discontinued therapy, with the majority ($n$ = 24) owing to progressive disease (PD). Only 23 patients reached the first response assessment time point, and in the monotherapy WM cohort, only one of four evaluable patients had a minor response at the first dose level (80 mg) at the last data cut-off.

Another novel compound, lisaftoclax (APG-2575), is a selective orally bioavailable BCL-2 inhibitor. In preclinical xenograft studies, after lisaftoclax treatment, predeath proteins, including Noxa and BCL-2-line protein 11 (BIM) were found to be increased [58]. The first-in-human phase 1 results of lisaftoclax were presented at the American Society of Clinical Oncology (ASCO) 2021 Meeting. In 12 of 14 evaluable patients with CLL/SLL, an ORR of 86% was observed [59]. However, in the non-CLL group, no partial responses were attained. Among 35 patients with R/R hematological malignancies, including four patients with WM, no dose-limiting toxicities were reported and a dose of up to 1200 mg was reached. There was no laboratory or clinical tumor lysis syndrome observed, and the major grade 3–4 treatment-related AE were cytopenias (neutropenia: 14%, thrombocytopenia: 7%).

Similar to the arena of *BTKi*, the field of BCL-2 inhibitors is getting crowded and the recent premature closure of a phase 1 study evaluating a novel promising BCL-2 inhibitor, LOXO-338, in B cell malignancies, including WM was allegedly a result of a strategic business decision by the study sponsor [60].

## 5. Proteasome Inhibitors

Proteasome inhibitors are a class of commonly used drugs in multiple myeloma (MM) that have also been extensively evaluated in WM, and, more recently, been adopted into the management of patients with WM [61,62]. PIs block the ubiquitin-proteasome pathway leading to downstream dysregulation of multiple pathways, increased endoplasmic reticulum stress, and activation of apoptotic pathways. Bortezomib, ixazomib, oprozomib and carfilzomib are four PIs, each with different affinities for the proteasome subunits, and different degrees of reversibility, that have been evaluated in studies involving patients with WM (Table 1) [33,34].

### 5.1. Bortezomib

The first-in-class PI, bortezomib, was initially investigated by Dimopoulos et al. in 10 patients with previously treated WM [33]. The study found an ORR in 80%, with 60% of patients achieving PR. The therapy was well tolerated, but a significant number of patients experienced treatment-emergent peripheral neuropathy (30%) (grade 2: $n = 1$, grade 3: $n = 2$). Subsequently, Chen et al. investigated bortezomib in 27 R/R and TN patients [34]. The investigators found that 21 patients had a reduction of IgM levels of at least 25%, with 44% having at least a 50% reduction (PR). In terms of safety, 74% of patients developed new or worsening peripheral neuropathy (grade 3 $n = 5$, no grade 4), and grade 3 or 4 thrombocytopenia (30%) and neutropenia (19%) were frequently observed. Similar efficacy results were observed in additional studies, which led to the investigation of combination regimens with bortezomib. Combination bortezomib, dexamethasone, and rituximab (BDR) was investigated in 23 patients with untreated symptomatic WM where an ORR and MRR of 96% and 83% were observed, respectively [63]. Additionally, a multinational, phase 2 trial assessed BDR in 65 untreated patients [64]. Here, an ORR of 85% was observed. PR was reported in 65%, and 68% achieved MRR. The PFS was 42 months, and 3-year OS was found to be 82%. Investigating the combination of bortezomib with cyclophosphamide and dexamethasone (CyBorD) with and without the addition of rituximab, a retrospective analysis was conducted at the Dana-Farber Cancer Institute [65]. Among the fifteen patients found to be treated with this regimen in the study time period, 7 (47%) received concurrent rituximab with CyBorD. An ORR of 93% was found, with an MRR of 53%. Grade 3–4 toxicities requiring dose changes/delays included neuropathy (26%), cytopenia (20%), and bacteremia (7%). Despite the small cohort, this analysis did demonstrate that the regimen provides robust efficacy outcomes [65]. The combination of bortezomib with dexamethasone, rituximab and cyclophosphamide (B-DRC) has additionally been investigated in a European phase 2, multicenter study for frontline treatment of WM [66]. A total of 204 patients were randomized 1:1 to DRC or B-DRC. At the end of treatment, B-DRC resulted in an MRR of 79% and ORR of 91% compared to 69% and 87% with DRC. A 2-year estimated PFS was found to be 81% with B-DRC compared to 73% with DRC ($p = 0.32$). Similar rates of grade 3 or higher AEs occurred in both arms (B-DRC 48%, DRC 47%), most commonly being neutropenia (25%), anemia (6%) and thrombocytopenia (5%) [66].

### 5.2. Carfilzomib

Carfilzomib is a second-generation PI that has been associated with a reduced risk of neuropathy, but an increased risk for cardiopulmonary toxicity [67–69]. After its approval for R/R MM, trials in WM were conducted. In a phase 2 trial examining a potentially neuropathy-sparing triplet comprising carfilzomib, rituximab and dexamethasone (CaRD) in 33 symptomatic untreated patients with WM, an ORR and MRR of 87% and 68% were observed, respectively [35]. The CR/VGPR rate was 36%. There was no grade 3 or higher treatment-related neuropathy, but grade 2 or higher hyperlipasemia (42%), neutropenia (14%) and cardiomyopathy (3%) was observed. Carfilzomib is also currently being assessed in combination with ibrutinib (NCT04263480), but has not been widely used in WM, with limited data outside of clinical trials [70].

### 5.3. Ixazomib

Ixazomib is an oral PI that blocks the activity of the 20S proteasome [69,71,72]. In a phase 2 study of 26 patients with TN WM, the combination of ixazomib, dexamethasone and rituximab resulted in an ORR of 96%, VGPR of 15%, and PR of 62%, with no CRs observed [36,73]. The therapy was well tolerated with common grade 2 or higher AEs being rash (8%) and insomnia. Combining fixed-duration ixazomib with subcutaneous flat-dose rituximab (following the first intravenous dosing) and dexamethasone, the phase 1/2 HOVON124/ECWM-R2 study investigated 59 patients with R/R WM [74]. Here, an ORR of 71%, VGPR of 14%, and PR of 37% was observed after eight cycles of therapy. The median PFS and OS were not reached. Safety analysis revealed that cycle delays due to hematological toxicity ($n = 6$), infusion-related reactions to rituximab ($n = 2$) neurotoxicity ($n = 5$) and other toxicities ($n = 21$) were commonly observed. Assessing the combination of ibrutinib with ixazomib, data from a phase 2 (NCT03506373), single-arm, trial were recently presented at ASH 2022 [75]. Of 21 patients analyzed with R/R and TN WM, an ORR of 76% with VGPR of 24% and PR of 52% was observed. The median time to progression was found to be 26 months. The safety analysis revealed anemia (81%), fatigue (76%), nausea (67%) and thrombocytopenia (52%) as the most common AEs.

### 5.4. Oprozomib

Oprozomib, another oral PI, was investigated in a phase 1b/2 study in 71 patients with WM and MM [37]. Two different oprozomib dosing schedules were utilized for the phase 2 component: once daily on days 1, 2, 8 and 9 (2/7 schedule) or days 1 to 5 (5/14 schedule) of a 14-day cycle. For patients with WM ($n = 14$) on the 2/7 schedule, the ORR was 71% with an MRR of 50% and 21% achieving a VGPR. On the 5/14 ($n = 17$) schedule, the ORR was much lower at 47% with an MRR of 29% and 0 patients achieving VGPR. Overall, no CRs were observed. Therapy was however limited by grade 3 or greater gastrointestinal and hematologic events, with a significant number of patients discontinuing due to AEs. Further evaluation of oprozomib in WM is not being pursued considering the treatment-emergent AEs.

Additional trials of oral and newer-generation PIs and combination regimens with *BTKi* in the frontline setting will provide interesting data, especially if the rates of CR or VGPR can be improved.

## 6. Phosphatidylinositol 3-Kinase Inhibitors

The phosphatidylinositol 3-kinase (PI3K) pathway plays an important role in cell survival and proliferation, and several PI3K inhibitors (PI3Ki) have been previously investigated in WM (Table 1) [76–79]. Idelalisib is a potent and highly selective PI3k/AKT inhibitor that was initially investigated in a phase 1b dose escalation study in patients with R/R B cell malignancies, including nine patients with WM [80]. Here, an ORR of 55% was observed; however, grade 3 transaminase elevation occurred in about 25% of patients [80]. Subsequently, idelalisib was also investigated in combination with obinutuzumab, a next-generation anti-CD-20 monoclonal antibody, in patients with R/R WM (NCT02962401) [38]. In this phase 2 study, the patients received idelalisib and obinutuzumab for six cycles during the induction phase ($n = 48$), followed by maintenance ($n = 27$) with idelalisib alone for 2 years or less [38]. Assessing the efficacy, five patients achieved VGPR, 27 achieved PR, and three patients had a minor response. Collectively, the ORR was 71%, with an MRR of 65%. Safety data was concerning as 26 patients were removed from the study due to AEs including neutropenia (9%), diarrhea (9%), and liver toxicity (9%). Hepatotoxicity with idelalisib was a major concern limiting its utility, and given the concerning safety profile, the study was prematurely closed. Other PI3 kinase inhibitors currently under investigation include duvelisib (NCT01882803) and umbralisib (NCT03364231) which may reveal a more favorable toxicity profile [81].

## 7. CAR-T CD19/CD20

Chimeric antigen receptor (CAR) T-cell therapy has revolutionized the care of relapsed/refractory B-cell malignancies and is currently FDA-approved for the treatment of acute lymphoblastic leukemia, certain non-Hodgkin lymphomas, and multiple myeloma [82–91]. Anti-CD19-CAR-T (CART-19) products were among the first constructs to be studied and approved in certain NHLs. WM cells express both CD19 and CD20, both of which are being explored as CAR-T cell target antigens [92]. Recently, Palomba et al. published data on three patients with WM, treated with CART-19, with two patients treated on NCT00466531 and one patient treated on NCT03085173 [93]. Varying degrees of responses were observed in all three patients (CR: 1 patient, PR: 1 patient, SD: 1 patient). However, disappointingly all patients relapsed between 3 and 26 months after the CAR-T therapy. Grade 1–2 cytokine release syndrome (CRS) was reported in all patients, with one patient exhibiting grade 2 neurotoxicity as well.

MB-106 is a CD20-targeted CAR that is being investigated in a phase 1/2 study in patients with R/R non-Hodgkin lymphoma (NCT03277729) [94]. Recently, Shadman et al. presented promising results from the trial at the 11th International Workshop on Waldenstrom Macroglobulinemia (IWWM-11) in Madrid, Spain. In 2 patients with WM who were treated so far, a 100% ORR was noted with both patients having persistence of CAR-T cells after infusion. One patient remained in remission at 15 months after treatment, while the other patient died from complications of COVID-19, 6 months after treatment in the absence of disease progression. Both patients experienced grade 1 CRS, with one patient experiencing immune effector cell-associated neurotoxicity syndrome (ICANS) as well. Further investigation of MB-106 assessing its safety and efficacy in WM will be of interest.

## 8. Bispecific Agents

Bispecific agents are antibodies with two binding sites directed against two different antigens or two different epitopes on the same antigen [95,96]. Generally, bispecific agents target one antigen on the tumor cell, and another antigen on immune effector cells, such as T cells. There is very limited data assessing the role of bispecific antibodies in WM. One such agent odronextamab (NCT03888105) is being studied in patients with R/R B-cell non-Hodgkin lymphomas, including WM [97]. Odronextamab is a hinge stabilized fully human IgG4-based anti-CD-20 X anti-CD3 bispecific which engages both the CD20+ WM cell (CD20) and the host T-cell (CD3) resulting in T-cell mediated cytotoxicity. The phase 1 results demonstrated an ORR of 51% among 145 heavily pretreated patients with R/R CD20-positive B cell malignancies; however, only 1 patient with WM has been reported to receive treatment so far [97]. The patients with follicular lymphoma who had received a dose of 5 mg or higher demonstrated an ORR of 91%, with a CR rate of 72%. Therapy was well tolerated, with the most common grade 3 or higher TRAEs being anemia (25%), lymphopenia (19%), hypophosphatemia (19%), neutropenia (19%), and thrombocytopenia (14%). Cytokine release syndrome occurred in 28% of patients. Plamotamab (XmAb 13676) is another anti-CD-20 × anti-CD3 bispecific antibody currently under investigation (NCT02924402) in CD20-expressing hematologic malignancies [98]. A case report of a 54-year-old woman with ibrutinib-refractory WM harboring *CXCR4* mutation who treated with plamotamab was recently published [98]. It demonstrated initial clinical efficacy with regression of an anterolateral high tumor, but eventual progressive disease while on therapy due to downregulation of CD20 on WM cells. Grade 2 CRS was additionally reported [98]. Future results from this trial will provide useful information regarding the agent's utility. Overall, given that bispecific T cell engagers have shown promising results in other indolent lymphomas, further exploration of these agents in patients with WM is anticipated.

## 9. CLOVER-WaM-Lopfosine

Iopofosine I 131 is a first-in-class radiotherapeutic which utilizes a targeted small-molecule phospholipid ether (CLR 1404) that is covalently bound to an isotope iodine-131 (I-311) [99]. The compound is internalized by targeting lipid rafts in tumor cells. Intracellularly, I-131 results in double-stranded DNA breaks via B-emission leading to tumor cell apoptosis. Previously, a phase 1 study evaluated iopofosine in patients with advanced solid tumors [100]. The open-label, multicenter, phase 2 study (NCT02952508) evaluated iopofosine I-131 (CLR 131) in select B-cell malignancies (CLOVER). A favorable efficacy and safety profile in heavily pretreated patients has been observed thus far [99,101]. This has led to the pivotal expansion specifically evaluating patients with WM who have received at least two prior lines of therapy (CLOVER-WaM). The primary objective of the trial is to determine the proportion of patients with WM achieving an MRR. Results from this ongoing trial may provide a novel therapeutic strategy for patients with R/R WM, especially if it results in high rates of CR or is beneficial for patients with *CXCR4* mutation.

## 10. Anti-*CXCR4* Agents

*CXCR4* mutations are associated with reduced response to therapy and shorter PFS. Additionally, these mutations impact *BTK*-inhibitor responses and lengthen the time to first and best response. Inhibitors of *CXCR4* including ulocuplumab and mavorixafor have been developed to improve disease response and long-term outcomes. Ulocuplumab, a *CXCR4* antagonist, was investigated in combination with ibrutinib in a phase 1 trial in 13 symptomatic WM patients (nine TN, four R/R) with *CXCR4* mutation [102]. Short median times to minor (<1 month) and major responses (1.2 months) were observed. A major and VGPR rate of 100% and 33% were observed, respectively. A median 2-year PFS of 90% was reported, and the most common grade $\geq$2 AEs included reversible thrombocytopenia (*n* = 2), rash (*n* = 5) and skin infections (*n* = 4). These remarkable results appear to favorably compare against ibrutinib monotherapy in this harder to treat patient subset, but further development of this intravenously administered drug was discontinued in WM. Mavorixafor is an oral small-molecule inhibitor of *CXCR4* which is under investigation in a phase 1b, open-label, multicenter single-arm study (NCT04274738) assessing the combination of mavorixafor with ibrutinib exclusively among patients with confirmed *MYD88* plus *CXCR4*$^{\text{WHIM}}$ mutations. The initial results presented at ASH 2021 demonstrated that among 10 patients enrolled, an ORR of 100% was observed in evaluable patients, with all patients demonstrating a rapid reduction in serum IgM on treatment [103]. Doses of 200 mg and 400 mg of mavorixafor were deemed to be safe in combination with ibrutinib and most AEs observed were of grade 1 (79%). Future trial results assessing these anti-*CXCR4* agents will be of interest given their early clinical efficacy demonstrated in the aforementioned studies.

## 11. Other Targeted Therapies

A host of additional targeted agents have been investigated in WM. Those worthy of discussion have included daratumumab, an anti-CD38 monoclonal antibody that is approved in the treatment of patients with MM. After preclinical data demonstrated promising efficacy, a multicenter phase 2 study assessed daratumumab monotherapy in 13 R/R patients [104,105]. A modest activity, with an ORR of 23% and MRR of 15%, translating to a median PFS of only 2 months, was observed. Grade 3 or 4 AEs included thrombocytopenia, neutropenia, bacteremia, increased alanine aminotransferase, and lymphopenia and the study was prematurely closed [105].

Everolimus is an inhibitor of mTOR, a serine-threonine kinase downstream of the PI3K/AKT pathway. In 50 patients with R/R WM treated with everolimus, a phase 2 trial demonstrated an ORR of 70%, with 42% achieving a PR and 28% having minimal response. The most prevalent grade 3 and 4 toxicities were anemia (18%), leukopenia (20%), thrombocytopenia (16%) and neutropenia (14%) [106]. In a trial investigation of 33 previously TN WM patients, the ORR with everolimus was 73%, with an MRR of 61% [107]. The authors found that discontinuation of therapy led to rapid serum IgM rebound in 7 patients, and symptomatic hyperviscosity in two patients. The most common AEs included mucositis (27%), infection (21%), and rash (21%), with the most common grade 2 or greater toxicities being anemia (27%), neutropenia (18%) and thrombocytopenia (15%). Despite promising preclinical data, the concerning toxicities encountered with everolimus pose a major barrier to its routine use. Selinexor is an orally bioavailable selective inhibitor of exportin 1 (XPO1), a nuclear export protein responsible for nuclear export of cargo proteins, including tumor suppressor proteins. A multicenter, phase 1 study (NCT01607892) assessed selinexor in 81 patients with R/R MM and three patients with R/R WM [108]. In all 84 patients, an ORR of 10% was observed. Additionally, the most common grade 3/4 toxicities included thrombocytopenia (45%), hyponatremia (26%) and anemia (23%). Patients with WM were included in the dose-escalation phase of the trial. However, the response data were not reported separately from the MM patients entered on the trial, thus, the benefit they derived from therapy is unclear [108]. Another therapy under investigation is loncastuximab terisine (ADCT-401), an antibody-drug conjugate composed of a humanized anti-CD19 monoclonal antibody that is attached to a pyrrolobenzodiazepine (PBD) dimer toxin (SG3199). A phase 1 study investigated 183 patients with R/R B cell malignancies, including one patient with WM [109]. Here, the ORR in evaluable patients was 46% with 27% of patients having CR. Safety analysis demonstrated fatigue (43%), nausea (32%), peripheral edema (32%) and GGT elevation (31%) to be the most common AEs. Grade 3 or higher TRAEs occurred in 77% of patients, with neutropenia (40%), thrombocytopenia (27%) and GGT elevation (21%) being most prevalent. The recommended dose was established (50 µg/kg once every 3 weeks for two cycles, followed by 75 µg/kg once every 3 weeks), and a phase 2 trial is underway, specifically assessing patients with WM (NCT05190705).

Perifosine is an Akt inhibitor which demonstrated efficacy in preclinical studies. Assessing the agent in 37 patients with R/R WM in a phase 2 trial (NCT00422656), a dose of oral perifosine at 150 mg daily was administered for 6 cycles and patients with stable or responding disease were allowed to continue until progression [110]. Here, a median PFS of 12.6 months was observed, with 11% of patients achieving a PR, 24% achieving a minimal response, and 54% had stable disease. AEs associated with therapy included cytopenias (grade 3–4, 13%), gastrointestinal symptoms (grade 1–2, 81%) and arthritis flare (all grade, 11%) [110]. Belimumab is a monoclonal antibody which inhibits the protein *BLYS*, which is involved in B-cell proliferation and inhibition of apoptosis. However, a phase 2 study that assessed belimumab in 12 patients with WM, disappointingly showed no meaningful change in serial serum IgM, and no objective responses were observed in all 12 patients [111].

The PembroWM study is a UK-wide phase II investigation (NCT03630042) assessing the combination of checkpoint inhibitor, pembrolizumab, and rituximab in patients with R/R WM. Results from this trial were recently presented at ASH 2022, where among 17 patients registered, at 52 weeks post-treatment, an ORR of 36% (VGPR: 9%, PR: 27%) was reported [112]. A median PFS of 12.6 months was observed and median OS was not reached. The most common AEs included infusion-related reactions (35%), anemia (29%), and fever (29%). Further results assessing the utility of PD-1 blockade in combination regimens for patients with R/R WM will be useful.

## 12. Challenges of Drug Development and Failed Trials

Certainly, while several promising novel targeted therapies have emerged, this has not been without several challenges and failed drug trials. Much of the difficulties in translating pre-clinical data into clinical response relies upon the ability to mimic the complex tumor microenvironment in animal models. Given that the animal models cannot reliably predict human response to therapy, preclinical data should be evaluated with a careful lens. Additionally, although a novel therapy may only demonstrate modest results in a large population of patients, certain subsets of patients with predictive markers or mutations which the drug targets may benefit the most. Thus, the selection of patients and enrolment of those that may derive the most clinical benefit based on certain disease markers should be employed to a greater degree [113]. As a consequence of some of these concerns, many trials in WM have failed.

These trials include NCT0060294, which set to investigate the radioimmunotherapy yttrium Y 90 ibritumomab tiuxetan in combination with rituximab for the treatment of WM. The study was terminated due to no accrual. Another trial was designed to assess epratuzumab, a humanized monoclonal antibody which binds to glycoprotein CD22 on B-cells (NCT00113802). However, this was terminated due to low accrual. IMO-8400 is an oligonucleotide antagonist of endosomal Toll-like receptors 7, 8, 9, that was investigated in a phase 1/2 clinical trial in patients with R/R WM (NCT02092909). Preliminary results were presented at ASH 2015, where of 17 patients enrolled, therapy was found to be well tolerated, apart from transient flu-like symptoms, and initial evidence of clinical activity was reported [114]. However, this trial was later terminated given the lack of efficacy. Alemtuzumab (Campath-1H), an anti-CD52 antibody was assessed in 28 symptomatic patients, 27 patients with WM and one patient with IgA-secreting LPL [115]. An overall and major response rate of 75% and 36% was observed. Complications included CMV reactivation (18%), new-onset autoimmune thrombocytopenia (14%), and grade 3 or higher hematologic toxicities including neutropenia (54%), thrombocytopenia (25%) and anemia (11%) and the use of this markedly toxic agent has been abandoned.

There are additional examples of poor translation of drug efficacy, impaired safety or low accrual which have resulted in failure of trials. As research continues in the field of WM, considerable emphasis to be given to international collaborative efforts to expeditiously bring trials studying this rare malignancy to fruition, but simultaneously the regulatory bodies should continue to apply stringent regulations on the approval of novel therapies for early trials. Only those promising agents which have a clear benefit over the currently available therapies should be offered to patients.

## 13. Our Approach in the Era of Targeted Therapies for WM

Despite the ease of administration, given the requirement for using *BTKi* continuously until intolerable AEs or disease progression, as well as their somewhat similar efficacy in the frontline versus relapsed/refractory settings, we use fixed-duration bendamustine-rituximab regimen in the frontline setting and reserve a *BTKi* as the first salvage therapy. However, this approach is advocated in the absence of high-level evidence through randomized controlled trials, comparing the two vastly different approaches. If a patient prefers to receive an oral non-chemo immunotherapy, we suggest using zanubrutinib over ibrutinib owing to a largely more favorable toxicity profile as well as somewhat greater efficacy of the former agent. Moreover, it is found to be effective in both $MYD88^{\text{L265P}}$ and $MYD88^{\text{WT}}$ WM. We reserve the use of the commercially available BCL2 inhibitor, venetoclax, and reversible, non-covalent *BTKi*, pirtobrutinib following second or higher relapse. Everolimus, an mTOR inhibitor, is best relegated to the multiply relapsed patient population, provided effective and less toxic approaches are unavailable.

## 14. Conclusions

Several novel therapies against WM have demonstrated promising efficacy along with a manageable toxicity profile. Remarkable results from clinical trials involving *BTKi* and other targeted therapies such as PIs and BCL-2 inhibitors have led to the frequent utilization of these agents from a variety of drug classes in the care of patients with WM. Further examination of novel approaches including CAR-T, bispecific antibodies, and radiotherapeutics, has the potential to change the paradigm of treatment in the R/R setting as we await evidence from ongoing larger studies. Exploration of rationale combination and sequential strategies with newer-generation agents that lack overlapping toxicities as well as the identification of novel 'druggable' targets is bound to further advance the field.

**Author Contributions:** Both K.L.C. and P.K. analyzed the literature, prepared the manuscript, and approved it in its final form. All authors have read and agreed to the published version of the manuscript.

**Funding:** There was no funding received for this work.

**Institutional Review Board Statement:** Not applicable.

**Informed Consent Statement:** Not applicable.

**Data Availability Statement:** Not applicable.

**Acknowledgments:** Figures were created with BioRender.com.

**Conflicts of Interest:** Karan L. Chohan declares no COI. Prashant Kapoor, FACP has received research funding from Amgen, Regeneron, BMS, Loxo Pharmaceuticals, Ichnos, Karyopharm, Sanofi, AbbVie, Angitiabio, GSK, and has provided consultancy or served on the advisory boards of BeiGene, Pharmacyclics, X4 Pharmaceuticals, Oncopeptides, Angitia Bio, GSK, AbbVie and Sanofi.

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
