# Peer review of "BTK Inhibitors and Other Targeted Therapies in Waldenström Macroglobulinemia"

_hemato, doi:10.3390/hemato4020012_

Round 1
Reviewer 1 Report
Greetings
I read your manuscript on targeted therapies in Waldenström Macroglobulinemia with great interest. Overall, I believe the manuscript is promising but requires minor revisions before publication. Below, I provide a summary of my review.
Introduction
I find the introduction already provide adequate background and information relevant to the study
Review
While the manuscript provides a balanced summary, there are some issues that need to be addressed. Firstly, many paragraphs are too lengthy and should be divided into multiple paragraphs based on the topic. Secondly, I recommend adding figures such as a comparison of non-covalent and covalent BTKi and a summary figure of all targeted therapies (e.g. BTKi, Proteasome Inhibitors, BCL-2 inhibitors, etc.). These figures can help clarify the information presented in the manuscript.
Additionally, some important aspects, such as the resistance of covalent BTKi and its side effects, require further elaboration. The mechanisms underlying the side effects of BTKi are also superficially addressed and could benefit from a more detailed explanation.
The table of “Select key clinical trials of targeted therapies in the treatment of WM” should also incorporate side effects data. Also, kindly separate overall survival and progression free survival into 2 different columns instead of combining the 2 outcomes into one column.
Another issue that I observe is that many sentences lack references or the references are not put within the same sentence such as:
- Unfortunately, these covalent agents can be limited by their requirement for daily continuous use, intolerance caused by off-target inhibition of other kinases, acquisition of mutations, including BTKCys481 mutations that may lead to disease progression.
- Ibrutinib, the first-in-class covalent BTKi, was approved in 2015 after the pivotal trial by Treon et al. revealed high rates of clinical response and improvement in anemia in the R/R setting
- A case report of a 54-year-old woman with ibrutinib-refractory WM harboring CXCR4 mutation who treated with plamotamab was recently published.
- The recent premature closure of a phase 1 study evaluating a novel promising BCL-2 inhibitor, LOXO-338, in B cell malignancies, including WM was allegedly a result of a strategic business decision by the study sponsor.
It appears that the list of references used in the review article are appropriate. In addition, there does not seem to be any issue with self-citation as only one self-cited reference is found in the manuscript (Kapoor P, Gertz MA, Laplant B, et al. Phase 2 Trial of Daratumumab, Ixazomib, Lenalidomide and Modified Dose Dexamethasone in Patients with Newly Diagnosed Multiple Myeloma. Blood. 2019;134(Supplement_1):864-864.)
I commend and appreciate the authors for their diligent work and believe that they are capable of revising the manuscript to meet the necessary standards for publication. I wish them all the best in their future endeavors.
Author Response
Greetings
Comment: I read your manuscript on targeted therapies in Waldenström Macroglobulinemia with great interest. Overall, I believe the manuscript is promising but requires minor revisions before publication. Below, I provide a summary of my review.
Introduction
I find the introduction already provide adequate background and information relevant to the study.
Review
While the manuscript provides a balanced summary, there are some issues that need to be addressed. Firstly, many paragraphs are too lengthy and should be divided into multiple paragraphs based on the topic. Secondly, I recommend adding figures such as a comparison of non-covalent and covalent BTKi and a summary figure of all targeted therapies (e.g. BTKi, Proteasome Inhibitors, BCL-2 inhibitors, etc.). These figures can help clarify the information presented in the manuscript.
Response: We thank the reviewer for their assessment of our manuscript. As suggested, we have divided the review into paragraphs based on topics and included further subheadings. Additionally, we have created two figures which compare both covalent and non-covalent BTKi therapies and demonstrate novel agents for the treatment of WM.
Comment: Additionally, some important aspects, such as the resistance of covalent BTKi and its side effects, require further elaboration. The mechanisms underlying the side effects of BTKi are also superficially addressed and could benefit from a more detailed explanation.
Response: We thank the reviewer for pointing this out. Accordingly, we have added a more detailed explanation of the resistance of covalent BTKi, side effects and the underlying mechanisms (Page 4, Line 10).
Comment: The table of “Select key clinical trials of targeted therapies in the treatment of WM” should also incorporate side effects data. Also, kindly separate overall survival and progression free survival into 2 different columns instead of combining the 2 outcomes into one column.
Response: We have updated Table 1 according to this suggestion and included side effect data.
Comment: Another issue that I observe is that many sentences lack references or the references are not put within the same sentence such as:
- Unfortunately, these covalent agents can be limited by their requirement for daily continuous use, intolerance caused by off-target inhibition of other kinases, acquisition of mutations, including BTKCys481 mutations that may lead to disease progression.
- Ibrutinib, the first-in-class covalent BTKi, was approved in 2015 after the pivotal trial by Treon et al. revealed high rates of clinical response and improvement in anemia in the R/R setting
- A case report of a 54-year-old woman with ibrutinib-refractory WM harboring CXCR4 mutation who treated with plamotamab was recently published.
- The recent premature closure of a phase 1 study evaluating a novel promising BCL-2 inhibitor, LOXO-338, in B cell malignancies, including WM was allegedly a result of a strategic business decision by the study sponsor.
It appears that the list of references used in the review article are appropriate. In addition, there does not seem to be any issue with self-citation as only one self-cited reference is found in the manuscript (Kapoor P, Gertz MA, Laplant B, et al. Phase 2 Trial of Daratumumab, Ixazomib, Lenalidomide and Modified Dose Dexamethasone in Patients with Newly Diagnosed Multiple Myeloma. Blood. 2019;134(Supplement_1):864-864.)
Response: Thank you for raising this important point. We have reviewed all the sentences in the manuscript and updated the references accordingly.
Comment: I commend and appreciate the authors for their diligent work and believe that they are capable of revising the manuscript to meet the necessary standards for publication. I wish them all the best in their future endeavors.
Response: We appreciate the reviewer’s thoughtful comments to improve the quality of the manuscript.
Reviewer 2 Report
In their manuscript the authors provided a comprehensive and updated review on the current therapeutic options, including also novel agents under investigation, for the patients with Waldenström Macroglobulinemia (WM). All classes of drugs are extensively explored so that a complete picture of the WM treatment choices can be obtained.
A recommendation is to include in the Introduction a brief description of the genetic landscape in WM. Also, a figure to summarize the main therapeutic options and their molecular targets would add more clarity to the written material.
Author Response
Comment: In their manuscript the authors provided a comprehensive and updated review on the current therapeutic options, including also novel agents under investigation, for the patients with Waldenström Macroglobulinemia (WM). All classes of drugs are extensively explored so that a complete picture of the WM treatment choices can be obtained.
A recommendation is to include in the Introduction a brief description of the genetic landscape in WM. Also, a figure to summarize the main therapeutic options and their molecular targets would add more clarity to the written material.
Response: We thank the reviewer for their assessment of our work and their useful feedback. As suggested, we have added a description of the genetic landscape of WM (Page 3, Line 7). Additionally, we have created a figure (Figure 1) highlighting the main therapeutic options and molecular targets of novel agents for the treatment of WM.